# Rab Geranylgeranyltransferase Subunit Beta as a Potential Indicator to Assess the Progression of Amyotrophic Lateral Sclerosis

**DOI:** 10.3390/brainsci13111531

**Published:** 2023-10-30

**Authors:** Jing Yang, Cheng Xin, Jia Huo, Xin Li, Hui Dong, Qi Liu, Rui Li, Yaling Liu

**Affiliations:** 1Department of Neurology, The Second Hospital of Hebei Medical University, Shijiazhuang 050000, China; doctor____yang@126.com (J.Y.); xincheng1110@163.com (C.X.); huojia719@163.com (J.H.); lixindoc666@163.com (X.L.); dong_ii@tom.com (H.D.); saltcovenant@163.com (Q.L.); lirui@hb2h.com (R.L.); 2The Key Laboratory of Neurology, Hebei Medical University, Ministry of Education, Shijiazhuang 050000, China; 3Neurological Laboratory of Hebei Province, Shijiazhuang 050000, China

**Keywords:** amyotrophic lateral sclerosis (ALS), monocyte, macrophages, Rab geranylgeranyltransferase subunit beta (RABGGTB)

## Abstract

Background: Currently, there is no effective treatment for amyotrophic lateral sclerosis (ALS), a devastating neurodegenerative disorder. Many biomarkers have been proposed, but because ALS is a clinically heterogeneous disease with an unclear etiology, biomarker discovery for ALS has been challenging due to the lack of specificity of these biomarkers. In recent years, the role of autophagy in the development and treatment of ALS has become a research hotspot. In our previous studies, we found that the expression of RabGGTase (low RABGGTB expression and no change in RABGGTA) is lower in the lumbar and thoracic regions of spinal cord motoneurons in SOD1G93A mice compared with WT (wild-type) mice groups, and upregulation of RABGGTB promoted prenylation modification of Rab7, which promoted autophagy to protect neurons by degrading SOD1. Given that RabGGTase is associated with autophagy and autophagy is associated with inflammation, and based on the above findings, since peripheral blood mononuclear cells are readily available from patients with ALS, we proposed to investigate the expression of RabGGTase in peripheral inflammatory cells. Methods: Information and venous blood were collected from 86 patients diagnosed with ALS between January 2021 and August 2023. Flow cytometry was used to detect the expression of RABGGTB in monocytes from peripheral blood samples collected from patients with ALS and healthy controls. Extracted peripheral blood mononuclear cells (PBMCs) were differentiated in vitro into macrophages, and then the expression of RABGGTB was detected by immunofluorescence. RABGGTB levels in patients with ALS were analyzed to determine their impact on disease progression. Results: Using flow cytometry in monocytes and immunofluorescence in macrophages, we found that RABGGTB expression in the ALS group was significantly higher than in the control group. Age, sex, original location, disease course, C-reactive protein (CRP), and interleukin-6 (IL-6) did not correlate with the ALS functional rating scale—revised (ALSFRS-R), whereas the RABGGTB level was significantly correlated with the ALSFRS-R. In addition, multivariate analysis revealed a significant correlation between RABGGTB and ALSFRS-R score. Further analysis revealed a significant correlation between RABGGTB expression levels and disease progression levels (ΔFS). Conclusions: The RABGGTB level was significantly increased in patients with ALS compared with healthy controls. An elevated RABGGTB level in patients with ALS is associated with the rate of progression in ALS, suggesting that elevated RABGGTB levels in patients with ALS may serve as an indicator for tracking ALS progression.

## 1. Introduction

Amyotrophic lateral sclerosis (ALS) is a chronic, progressive, and fatal neurodegenerative disease with a worldwide incidence of approximately 5/100,000–7/100,000. Pathological alterations occur as motor neurons in the anterior horn cells of the spinal cord, the brain stem, and the cerebral cortex undergo degenerative changes. Patients typically succumb to generalized muscle weakness and respiratory failure, with an average survival time of 3–5 years [1]. ALS is a rare, clinically heterogeneous disease that can be difficult to recognize, especially in the early stages. ALS is often misdiagnosed as more common illnesses due to the lack of specific markers in the diagnostic criteria for ALS, delaying diagnosis. Several biomarkers, including neurofilaments [2,3,4,5,6], interleukin-6 (IL-6) [7,8], C-reactive protein (CRP) [5,9], cholesterol [10], creatinine [11,12,13], uric acid [14,15,16,17], ferritin [18], TDP43 [19,20], and albumin [21], have been proposed as prognostic factors for ALS progression monitoring, but many factors, including age, BMI, etc., may be confounding factors that influence these biomarkers and should therefore be taken into account when evaluating these biomarkers in patients. Several studies [6,22] have suggested that neurofilament be used to monitor disease progression in ALS as well as for a differential diagnosis, but it lacks specificity, as it is a non-specific biomarker for a variety of neurological disorders including AD [23], MS [24], and frontotemporal lobe dementia [25]. A study from JAMA in 2017 found that ALS patients with elevated serum CRP levels progressed more rapidly than patients with lower CRP levels, with researchers concluding that serum CRP can be used as a prognostic biomarker for ALS [9]. CRP, a widely used non-specific biomarker of inflammation, has been found to be elevated in serum CRP in a number of gynecological oncological diseases and can be used as a prognostic indicator, including for cervical, ovarian, endometrial, and vulvar cancers [26,27]. This biomarker lacks specificity, as it is a non-specific biomarker for ALS, so biomarker discovery has been challenging for ALS. Although Riluzole [28] and Edaravone [29] have been approved by the Food and Drug Administration (FDA), there is currently no cure or effective treatment for ALS.

Rab protein [30] regulates cell signal transduction, cell growth, and differentiation as its primary function. Rab resides primarily in the plasma membrane and organelle membranes and is evolutionarily highly conserved. It regulates intracellular vesicle transport and is present in almost all eukaryotes, and is strictly regulated. Rab7 is a member of the Ras superfamily and plays a key role in the late stage of autophagy. Therefore, the dysfunction of Rab protein is linked to multiple human diseases, including cancer ([30,31,32,33,34], infection [30,35,36], cardiac disease [37], and neurodegeneration [38,39,40,41,42,43]. In recent years, an increasing number of researchers [44,45,46,47,48,49,50] have focused on the role of Rab protein in the development of ALS.

In our previous studies, we found that prenylation of Rab7 was inhibited in the ALS model, and RabGGTase mediated prenylation modification of Rab7. Autophagy obstacles have been confirmed to be one of the early pathological events of ALS [51,52], and autophagy plays an important role in inflammation [53]. In our previous studies, we found that the expression of RabGGTase (low RABGGTB expression and no change in RABGGTA) is lower in the lumbar and thoracic regions of spinal cord motoneurons in SOD1G93A mice compared with WT (wild-type) mice groups, and autophagy defects could be ameliorated by modulating RABGGTB in neurons [54]. However, it was difficult to obtain neurons in patients with ALS. RabGGTase is composed of Rab geranylgeranyltransferase subunit alpha (RABGGTA) and Rab geranylgeranyltransferase subunit beta (RABGGTB) [55], and our previous experiments determined that the RABGGTB expression level was decreased in spinal cord motoneurons in SOD1G93A mice, and the onset of SOD1G93A mice was significantly delayed and their survival time was prolonged by intrathecal injection of adeno-associated virus 9 (AAV9) carrying the human single chain RABGGTB gene. Given that RabGGTase is associated with autophagy and autophagy is associated with inflammation, important questions include how the expression of RabGGTase is in the peripheral inflammatory cells in patients with ALS, and whether it is associated with disease progression rate.

In recent years, several studies have focused on the role of RabGGTase in diseases. For instance, Taheri et al. compared the expression of genes coding for the different subunits of proteins implicated in protein prenylation between patients with multiple sclerosis and healthy subjects, and found that RABGGTB was significantly downregulated in the peripheral blood, suggesting dysregulation of the protein prenylation pathway in MS [56], whereas high RABGGTB expression has been reported in tumor-associated disease, and psoromic acid (PA) as a selective Rab-prenylation inhibitor has been proposed to be potentially therapeutic for cancer and osteoporosis [57]. RABGGTB is not used as a biomarker in these diseases. Several biomarkers have been proposed as prognostic factors for monitoring ALS progression, but these biomarkers lack specificity, as they are non-specific biomarkers for ALS, so biomarker discovery has been challenging for ALS.

However, to the best of our knowledge, we are the first to demonstrate that RABGGTB was found to be highly expressed in peripheral mononuclear macrophages of ALS patients compared with healthy controls, and we found that the expression of RABGGTB was significantly correlated with disease progression levels (ΔFS). Increased RABGGTB in patients with ALS could be used as a biomarker for prognostic assessment of ALS.

## 2. Materials and Methods

### 2.1. Subjects

The ethics committee of the Second Hospital of Hebei Medical University (No. 2022-R196) approved this research. We collected medical histories, physical examination records, laboratory tests, and electrophysiological examination records, among other data sources, of patients diagnosed with ALS at the Second Hospital of Hebei Medical University’s Department of Neurology between January 2021 and August 2023. All patients satisfied the revised ALS diagnostic criteria [58]. The department of physical examination collected data on the corresponding healthy controls, including gender, age, height, weight, past history, family history, and laboratory examination, among other data. Exclusion criteria included patients with acute or chronic inflammatory diseases, such as acute pneumonia and rheumatoid arthritis. The consent forms were signed by all participants. The ALS functional rating scale—revised (ALSFRS-R) was used to evaluate the severity of the disease [59], and the rate of disease progression was evaluated using the ALSFRS-R score to calculate the progression rate ratio as follows: (48 − ALSFRS-R score at the time of diagnosis)/time from onset to diagnosis. Patients were divided into 3 groups according to ΔFS: slow (ΔFS < 0.5), intermediate (ΔFS = 0.5–1.0), and rapid (ΔFS > 1.0) [60,61].

### 2.2. Serum Sample

Venous blood was drawn from the elbow of patients with ALS and healthy controls in the morning after an overnight fast, and samples were collected in EDTA vacutainers, and immediately centrifuged for 15 min at ∼2000× *g* at room temperature. After centrifugation, plasma was removed, aliquoted into polypropylene tubes, and used for biochemical analyses (BECKMAN, AU5800, Brea, CA, USA). All blood indicators, including IL-6 and CRP, were supplied by the laboratory division of the Second Hospital of Hebei Medical University; 0–7 pg/mL was the normal range for adult IL-6, and 0–6 mg/L was the normal range for CRP.

### 2.3. Flowcytometry

After completing routine examinations, blood samples from patients with ALS and healthy controls were collected. Using flow cytometry, RABGGTB expression levels in peripheral blood mononuclear cells (PBMCs) were determined (BD Biosciences, Franklin Lakes, NJ, USA). Primary antibodies included were monoclonal mouse anti-human CD14 (BD, 555399), monoclonal mouse anti-human CD16 (BD, 560717), Fc block (BD, 564219), and anti-RABGGTB antibodies (1:200, GeneTex, Irvine, CA, USA; GTX105874). The secondary antibody was goat anti-rabbit IgG (H+L) conjugated with fluorescein-5-isothiocyanate (FITC) (Proteintech, Rosemont, IL, USA; SA00003-2).

### 2.4. Isolation of PBMCs from Blood Samples

Human peripheral blood lymphocyte separation liquid (LTS1077) was added to PBMCs (Tianjin Haoyang Biological Products Science & Technology Co., Ltd., Tianjin, China; 601002), placed in a high-efficiency centrifuge tube, and centrifuged at 200× *g* for 2 min at room temperature. The peripheral blood samples were then added and centrifuged for an additional 30 min at 800× *g*. The intermediate mononuclear cell layer was carefully aspirated into a new centrifuge tube and centrifuged at 300× *g* for 13 min. After aspirating the supernatant, the pellet was washed multiple times with phosphate-buffered saline (PBS), and the PBMCs were resuspended in a cryopreserved solution containing 90% fetal bovine serum (FBS; CellMax, Beijing, China, SA211.02) and 10% dimethyl sulfoxide (DMSO; Sigma-Aldrich, St Louis, MO, USA; D2650-100ML). The solution was then transferred to a 1.8 mL cryopreservation tube and stored at −80 °C overnight; it was then stored in liquid nitrogen for long-term preservation and examination at a later date.

### 2.5. In Vitro Culture of Macrophages

The previously prepared PBMCs were thawed in preheated Roswell Park Memorial Institute (RPMI)-1640 medium (Gibco, Waltham, MA, USA; C11875500BT) containing 10% FBS. The supernatant was removed after centrifugation at 500× *g* for 5 min at room temperature, and the cell pellet was resuspended in RPMI-1640 culture medium with 1% penicillin-streptomycin (P/S) and 10% FBS. The culture medium was supplemented with macrophage colony-stimulating factor (M-CSF) (PeproTech, East Windsor, NJ, USA; 300-25-10), and the cells were seeded in 48-well plates for 7 days. The culture medium was replaced every 3 days [61]. On the seventh day of cell culture, the medium was withdrawn and fixed with 4% paraformaldehyde in PBS.

### 2.6. Immunofluorescence and Confocal Microscopy Analysis

Using a PBS solution containing 4% paraformaldehyde, the cells were fixed on 48-well plate slides. The cells were then permeated with PBS containing 0.3% Triton-X 100 for 15 min and blocked for 1 h at room temperature with PBS containing 10% sheep serum. After adding the primary antibody and allowing it to incubate overnight, the cells were washed three times with PBS and then incubated with the corresponding secondary antibody: donkey anti-rat IgG (H+L) highly cross-adsorbed secondary antibody (CD68(1:500, Abcam, Waltham, MA, USA, ab31630), F4/80(1:200, Abcam, Waltham, MA, USA, ab6640), and RABGGTB (1:500, GeneTeX, San-Antonio, TX, USA, GTX105874)), Alexa Fluor 488 (1:1000; Invitrogen, Carlsbad, CA, USA; A21208), Alexa Fluor 594-conjugated goat anti-mouse secondary antibody (1:1000; Thermo Fisher Scientific, Waltham, MA, USA; #A-11032), and goat anti-rabbit secondary antibody labeled with Alexa Fluor 647 (1:1000, Invitrogen, A21245). After 1 h of staining at room temperature with 4′,6-diamidino-2-phenylindole (DAPI), the cell nuclei were visible. The cells were then washed three times with PBS and observed using a confocal fluorescence microscope (ZEISS, Oberkochen, Germany; LSM900). The parameters of the microscope were set at the start of each individual imaging process and remained constant throughout.

### 2.7. Statistical Analysis

All enumeration data are presented as mean ± standard deviation (SD), and the Shapiro–Wilk test was used to examine the data distribution. For data with a normal distribution, the unpaired *t*-test was used to analyze the differences between two groups, whereas the Pearson’s correlation coefficient test was utilized to analyze the correlation. The Mann–Whitney U test was used for analysis of differences, Spearman’s rank correlation coefficient test was used for correlation analysis, and the multivariable regression model was adjusted for age, sex, and body mass index (BMI). All statistical analyses were conducted using GraphPad Prism 9 (GraphPad Software Inc., San Diego, CA, USA) and SPSS26. *p* < 0.05 was considered statistically significant.

## 3. Results

### 3.1. General Information of the Participants

The data of 71 patients from 86 patients diagnosed with ALS at the Second Hospital of Hebei Medical University between January 2021 and August 2023 were collected (Figure 1). The healthy control group consisted of 54 subjects (male: 35, female: 19) with a mean age of 56 ± 8 years and BMI of 23.44 ± 1.839. The ALS group included a total of 71 patients (male: 47, female: 24), with a mean age of 58 ± 9 years. The distributions of age, gender, and BMI did not differ significantly (Table 1).

### 3.2. Elevated RABGGTB Levels Were Detected in the Monocytes from Patients with ALS

Flow cytometry was used to detect RABGGTB levels in monocytes from patients with ALS compared with healthy controls. Monocytes are important mononuclear blood cells that participate in the immune and inflammatory responses of humans. Cells were divided into three subpopulations based on the level of CD14 and CD16 surface expression: classical (CD14++CD16), intermediate (CD14++CD16+), and nonclassical (CD14+CD16+) monocyte subpopulations [62]. Comparing the expression levels of RABGGTB in classical-type monocytes between the groups, we discovered that RABGGTB was significantly upregulated in the peripheral blood of patients with ALS (Figure 2A–C).

### 3.3. Correlations between RABGGTB Levels in Monocytes with ALS Disease Severity or Progression Rates

The above findings suggest that RABGGTB expression may be associated with the clinical status of patients with ALS. We included indexes such as gender, age, BMI, site of disease onset, disease progression, CRP, IL-6, RABGGTB, and ALSFRS-R score. We first determined the relationship between disease severity and the aforementioned indicators. Gender, disease course, site of onset, BMI, CRP, and IL-6 did not correlate with ALSFRS-R score (Appendix A), whereas age and RABGGTB level were significantly correlated with ALSFRS-R score (*p* = 0.0418, *p* = 0.0441, respectively). Further multivariate analysis revealed a significant correlation between age and ALSFRS-R score (*p* = 0.0079). The results suggest that age and the expression of RABGGTB are associated with the severity of ALS (Figure 3A,B).

Then we evaluated the correlation between the indicators and the progression rate of the disease. Age, gender, site of onset, BMI, CRP, and IL-6 were not significantly correlated with ΔFS (Appendix A), whereas RABGGTB expression level and disease course were significantly correlated with ΔFS (*p* = 0.002, *p* = 0.0007, respectively). The course of disease and RABGGTB were significantly correlated with ΔFS (*p* = 0.001 and *p* = 0.025, respectively). The results indicate that the expression of RABGGTB and the course of disease are related to the progression rate of the disease—the stronger the expression of RABGGTB, the shorter the disease course, and the faster the disease progressed (Figure 3C,D).

### 3.4. Elevated RABGGTB Levels Were Detected in Monocyte-Derived Macrophages Derived from Patients with ALS

As we know, most tissue macrophages are derived from peripheral blood monocytes; therefore, we induced monocyte differentiation into macrophages to further assess the expression of RABGGTB in macrophages. The data of 47 patients from 62 patients diagnosed with ALS at the Second Hospital of Hebei Medical University between January 2021 and October 2022 were collected (Figure 4). The healthy control group consisted of 34 subjects (male: 20, female: 14) with a mean age of 54 ± 7 years and BMI of 23.53 ± 2.172. The ALS group included a total of 47 patients (male: 32, female: 15), with a mean age of 58 ± 10 years and BMI of 23.28 ± 2.463. The distributions of age, gender, and BMI did not differ significantly (Table 2).

After M-CSF stimulation, monocytes extracted from peripheral blood differentiated into macrophages. The cells displayed typical macrophage morphology, with enlarged cell volume, round or oval shape, and pseudopodia exhibiting typical amoeboid morphology (Figure 5A), with CD68 and F4/80 macrophage markers expressed on the cell surface (Figure 5B). Immunofluorescence semiquantitative results revealed that the expression of RABGGTB in macrophages was significantly higher (*p* < 0.01) in the ALS group (17.340 ± 8.226) than in the control group (5.671 ± 2.932) (Figure 5C,D).

### 3.5. Correlations between RABGGTB Levels in Monocyte-Derived Macrophages with ALS Disease Severity or Progression Rates

The above findings suggest that RABGGTB expression may be associated with the clinical status of patients with ALS. We included indexes such as gender, age, BMI, site of disease onset, disease progression, CRP, IL-6, RABGGTB, and ALSFRS-R score. We first determined the relationship between disease severity and the aforementioned indicators. Age, gender, disease course, site of onset, BMI, CRP, and IL-6 did not correlate with ALSFRS-R score (Appendix A), whereas RABGGTB level was significantly correlated with ALSFRS-R score (*p* = 0.007). Further multivariate analysis revealed a significant correlation between RABGGTB and ALSFRS-R score (*p* = 0.041). The results suggest that the expression of RABGGTB is associated with the severity of ALS. The severity of the disease increased with the intensity of RABGGTB expression (Figure 6A).

We also evaluated the correlation between the indicators and the progression rate of the disease. Age, gender, site of onset, BMI, CRP, and IL-6 were not significantly correlated with ΔFS (Appendix A), whereas RABGGTB expression level and disease course were significantly correlated with ΔFS (*p* < 0.01). The course of disease and RABGGTB were significantly correlated with ΔFS (*p* = 0.001 and *p* = 0.016, respectively). The results indicate that the expression of RABGGTB and the course of the disease are related to the progression rate of the disease—the stronger the expression of RABGGTB, the shorter the disease course, and the faster the disease progressed (Figure 6B,C).

## 4. Discussion

Amyotrophic lateral sclerosis (ALS) is a progressive fatal neurodegenerative disease, and the etiology and pathogenesis of ALS are unknown, leading to delayed diagnosis, so patients have a poor prognosis with an average survival time of 3–5 years [1]. There is growing evidence that monocytes, which are involved in the human immune response and inflammatory response, are implicated in ALS [63,64,65]. The expression levels of surface markers CD14 and CD16 were used to divide cells into three subgroups. Classical (CD14++CD16) monocytes participate in phagocytosis, immune response, and migration [66], which are essential for the body’s initial line of defense against innate immune defenses. Classical monocytes can differentiate into macrophages in tissues and mediate chronic diseases [67] such as obesity [68], Alzheimer’s disease [69], and cancer [70]. Nonclassical (CD14+CD16++) monocytes utilize the integrin lymphocyte function-associated antigen 1 (LFA-1) and the chemokine receptor CX3CR [71] to patrol the inner lining of vascular endothelium. Studies have demonstrated that nonclassical monocytes can differentiate into macrophages and participate in inflammatory responses under stimulation [72], and it has been suggested that nonclassical monocytes have a protective effect against chronic diseases and a positive correlation with disease burden [67]. There is more evidence demonstrating that monocytes in the peripheral circulation of patients with ALS promote inflammatory response [65]. According to our study, flow cytometry revealed that the RABGGTB expression levels of classical-type monocytes were significantly higher than those of controls, suggesting that RABGGTB may be linked to inflammation in classical monocytes.

To further demonstrate that the expression of RABGGTB was increased in patients with ALS, we extracted their monocytes. Using immunofluorescence, we detected the expression of RABGGTB in macrophages induced by monocytes from patients with ALS. Compared to the control group, the expression of RABGGTB was significantly higher in the ALS group. Macrophages [73]) play crucial roles in the onset and progression of ALS, in which their phenotypes and functions are altered. According to studies, M1 macrophages derived from monocytes in patients with ALS produce more proinflammatory cytokines than healthy controls [65]. Unfortunately, we did not further distinguish the phenotype of macrophages in this study; in subsequent research, we will examine the phenotype and function of macrophages.

We included gender, age, BMI, site of disease onset, disease progression, CRP, IL-6, and RABGGTB levels, and ALSFRS-R score based on the above results. In our analysis, the expression levels of RABGGTB were examined by flow cytometry and immunofluorescence, respectively, and the included indicators were used to perform correlation analyses. RABGGTB levels were not associated with sex, age at diagnosis, BMI, or disease onset site. ALSFRS-R score and ΔFS were also unrelated to gender, age at diagnosis, BMI, and site of onset. Multiple studies have confirmed that the gender disparity in onset of ALS is substantial [74,75]. There is evidence that estrogen has a protective effect in ALS, with ovariectomy hSOD1-G93A transgenic mice exhibiting earlier disease onset as well as diminished anti-inflammatory and anti-apoptotic abilities, suggesting that estrogen may play a significant role in protecting spinal motor neurons [76]. However, our analysis revealed no correlation, possibly because the majority of our patients were confined to the Hebei region, which may have contributed to these disparities. ALS is characterized by a gradual increase with age, with the incidence beginning to rise at 40 years and peaking between 60 and 70 years of age, followed by a steep decline [77]. Some age-specific studies compared the age distribution differences between continents and found that, with the exception of East Asia, the age-specific pattern of ALS incidence was consistent [77]. Multiple studies have determined that disease onset age is a significant factor in disease progression and prognosis [78]; the older the disease onset age is, the faster the disease progression rate and the shorter the survival [79]. In our analysis, we did not find a significant correlation between age and other variables. This may have been due to the limited sampling regions and small sample sizes.

Previous research has demonstrated that a higher BMI is associated with improved survival and that a high BMI and weight gain are associated with a lower risk of ALS [80]. RABGGTB levels were not found to be associated with BMI or disease onset site in our analysis. ALSFRS-R scores and ΔFS were also not associated with BMI or disease onset site. These differences may be attributable to sampling restrictions and the relatively small sample size, and we need to confirm our findings with larger samples.

A study published in JAMA found that patients with ALS with elevated serum CRP levels progressed more rapidly than those with lower CRP levels [9], suggesting that serum CRP could serve as a prognostic biomarker for ALS. A subsequent study analyzing the association between serum CRP and ALS concluded that CRP was not associated with ALS survival [5], suggesting that CRP is not a risk factor for ALS [81]. In our analysis, serum CRP was not associated with RABGGTB level, ALSFRS-R score, or ΔFS, indicating that serum CRP may be a useful prognostic biomarker for ALS; however, multicenter and multiregional validation are required. IL-6 is a multifunctional cytokine; IL-6 levels were significantly elevated in some patients, but not in the majority. Our analysis revealed that RABGGTB levels were not associated with serum IL-6, and based on our limited data, ALSFRS-R scores and ΔFS were also not associated with serum IL-6.

Decades of research on the disease course as a prognostic factor in ALS have revealed that the longer the delay between symptom onset and diagnosis [81], the better the prognosis. A disease with a shorter duration progresses more rapidly, necessitating more urgent treatment [79]. In our analysis, RABGGTB level and ALSFRS-R score were not associated with disease progression. However, ALSFRS-R score was associated with RABGGTB level, and ΔFS was associated with RABGGTB level and disease progression, regardless of age, gender, BMI, site of onset, or inflammation level. These findings suggest that the expression of RABGGTB is associated with the severity of ALS, and that patients with higher RABGGTB expression levels and a shorter disease course experience a more severe disease and aggressive progression. The significance of the RABGGTB level as a useful, feasible, and potential prognostic factor in patients with ALS is highlighted in this study.

RABGGTB as a potential indicator to assess the progression of ALS is not without limitations. The main limitation is that ALS is rare and heterogeneous in clinical manifestations, and the limited sample size means that our data are not fully representative of sporadic ALS (sALS), such as patients of different ethnicities or geographical regions. A second limitation is that the data collected come from sporadic ALS, and these data are not representative of familial ALS (fALS), such as those with mutations in the Cu/Zn-superoxide dismutase (SOD1) gene. The third limitation is that it is not clear what role RabGGTase plays in regulating inflammation and what effect inflammation has on RabGGTase. Therefore, we need a larger sample size to reliably comment on the value of RABGGTB as a prognostic biomarker. In subsequent experiments, we will investigate the role of RabGGTase in the regulation of inflammation and the effect of inflammation on RabGGTase. We will determine how RABGGTB responds in patients with different inflammatory conditions and whether the high expression is a primary phenomenon in macrophages or secondary to the inflammatory state of the cells.

## 5. Conclusions

In this study, we demonstrated for the first time that RABGGTB expression was greater in monocyte macrophages of patients with ALS than in healthy controls. Compared to controls, the expression of RABGGTB in monocytes and monocyte-differentiated macrophages derived from patients with ALS was correlated with progression rate—the stronger the expression of RABGGTB, the faster the disease progressed in patients with ALS, suggesting that RABGGTB may serve as an indicator for ALS progression.

## Figures and Tables

**Figure 1 brainsci-13-01531-f001:**
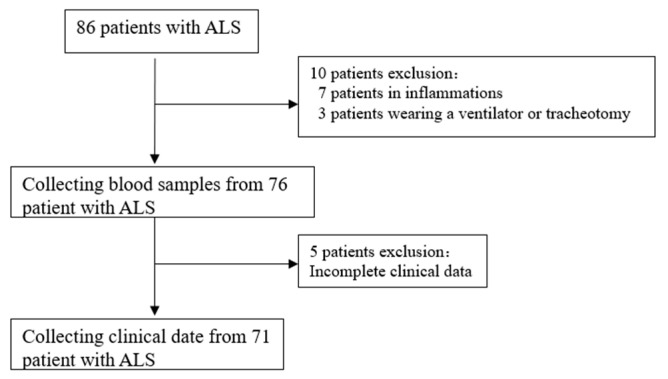
ALS patient screening. ALS, amyotrophic lateral sclerosis.

**Figure 2 brainsci-13-01531-f002:**
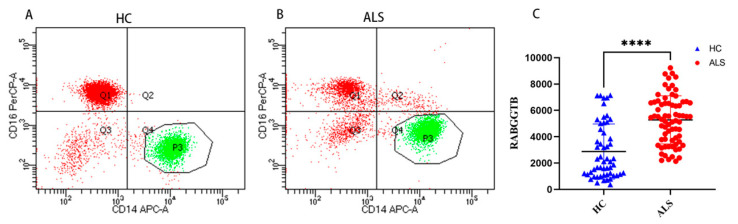
RABGGTB expression in monocytes from healthy control and patients with ALS. (**A**) Analysis of peripheral blood classical-type monocytes by flow cytometry in healthy controls. (**B**) Analysis of peripheral blood classical-type monocytes by flow cytometry in patients with ALS. (**C**) Quantitative analysis of the RABGGTB concentration in peripheral blood monocytes of classical type. The statistical significance was determined using an unpaired *t*-test. ****, *p* < 0.0001. HC, healthy control; ALS, amyotrophic lateral sclerosis; RABGGTB, Rab geranylgeranyltransferase subunit beta.

**Figure 3 brainsci-13-01531-f003:**
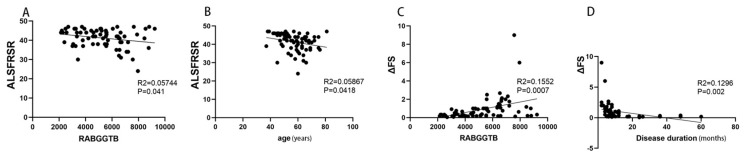
Correlation between ALSFRS-R score or ΔFS and RABGGTB expression in monocytes from ALS patients. (**A**) Correlation between ALSFRS-R and the expression of RABGGTB in monocytes from ALS patients. (**B**) Correlation between ALSFRS-R and age in ALS. (**C**) Correlation between ΔFS and the expression of RABGGTB in monocytes from ALS patients. (**D**) Correlation between ΔFS and the duration of the disease in ALS.

**Figure 4 brainsci-13-01531-f004:**
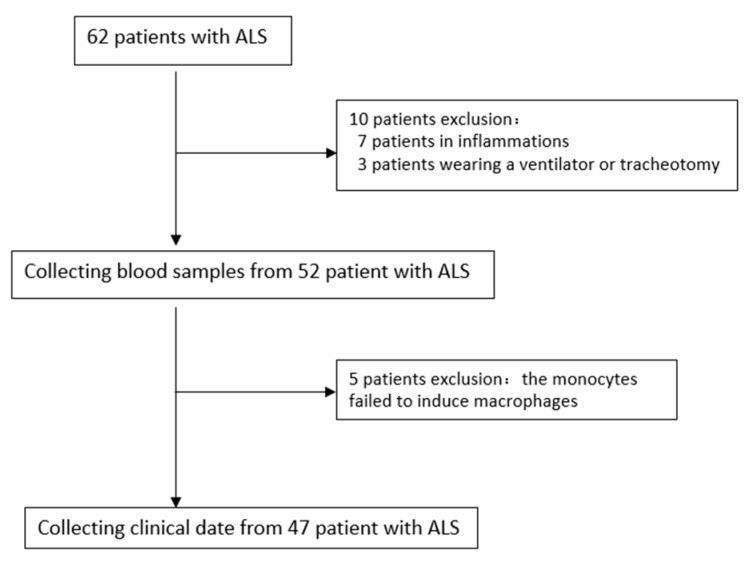
Selection of patients with ALS in this study. ALS, amyotrophic lateral sclerosis.

**Figure 5 brainsci-13-01531-f005:**
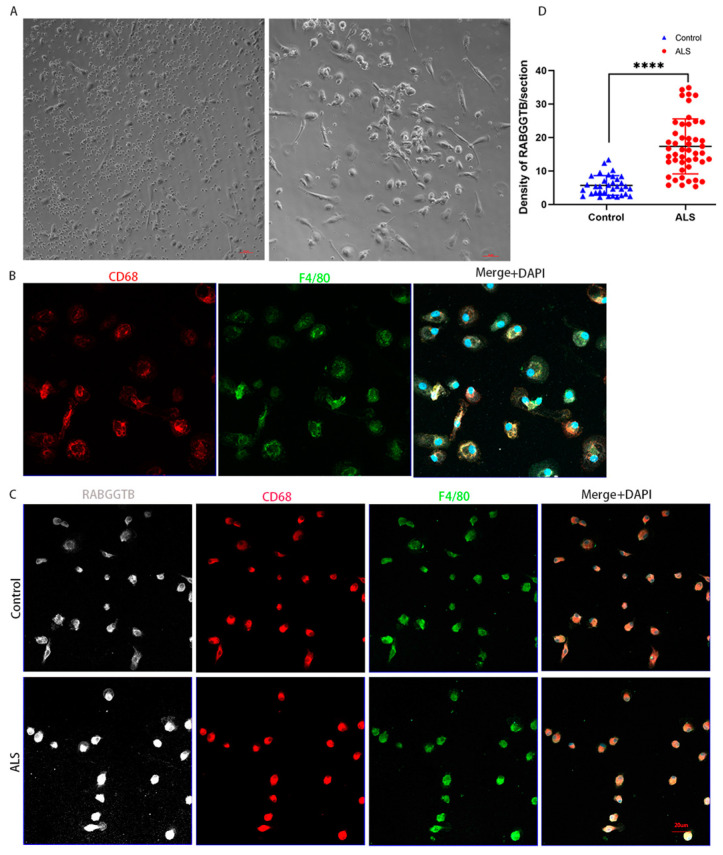
RABGGTB expression in macrophages derived from healthy control and patients with ALS. (**A**) Phase-contrast microscopy images of macrophages differentiated with 20 ng/mL M-CSF for 4 days. Scale bars = 50 µm. (**B**) Phase-contrast microscopy images of macrophages differentiated with 20 ng/mL M-CSF for seven days. Scale bars = 50 µm. Immunofluorescence labeling of monocyte-derived macrophages for CD68 (red) and F4/80 (green). DAPI was used to stain nuclei (blue). Scale bars = 20 µm. (**C**) Immunofluorescence labeling for RABGGTB (gray/white), CD68 (red), and F4/80 (green) in monocyte-derived macrophages from patients with ALS and healthy controls. DAPI was used to stain nuclei (blue). Scale bars = 20 µm. (**D**) Quantitative analysis of RABGGTB levels in monocyte-derived macrophages from patients with ALS and healthy controls. The statistical significance was determined using an unpaired *t*-test. ****, *p* < 0.0001. ALS, amyotrophic lateral sclerosis; RABGGTB, Rab geranylgeranyltransferase subunit beta; M-CSF, macrophage colony-stimulating factor.

**Figure 6 brainsci-13-01531-f006:**
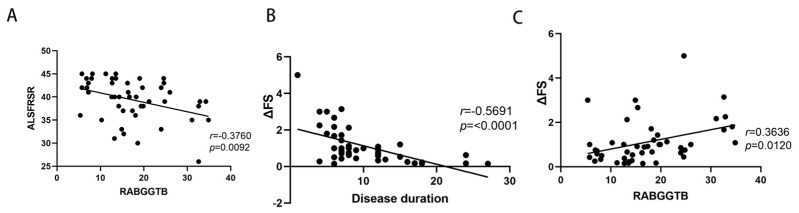
Correlation between ALSFRS-R score or ΔFS and RABGGTB expression in macrophages derived from monocytes. (**A**) ALSFRS-R is the revised ALS functional rating scale. (**B**) Correlation between ΔFS and the duration of the disease in ALS. (**C**) Correlation between ΔFS and the expression of RABGGTB in monocyte-derived macrophages.

**Table 1 brainsci-13-01531-t001:** Summary of donor information.

Variables	Healthy Controls	ALS Patients	*p* Value
Participants (number)	54	71	
Age (years)	56 ± 8	58 ± 9	0.3344
Sex (female/male)	19/35	24/47	
Site of onset			
Bulbar	NA	17	
Limb	NA	54	
BMI (kg/m^2^)	23.44 ± 1.839	22.64 ± 3.784	0.1568
Disease duration (months)	NA	13.92 ± 7.86	
CRP (mg/L)	NA	3.675 ± 5.356	
IL-6 (pg/mL)	NA	7.657 ± 6.564	
RABGGTB	2875 ± 2093	5280 ± 1826	<0.0001
ALSFRS-R score	NA	40.35 ± 4.859	
ΔFS	NA	1.092 ± 0.997	

BMI, body mass index; CRP, C-reactive protein; IL-6, interleukin 6; RABGGTB, Rab geranylgeranyltransferase subunit beta; ALSFRS-R, ALS functional rating scale—revised; ALS, amyotrophic lateral sclerosis; ΔFS, delta FS ratio (rate of disease progression) = (48 – ALSFRS-R score at time of diagnosis)/time onset to diagnosis); NA, not available.

**Table 2 brainsci-13-01531-t002:** Demographic parameters of healthy controls and ALS patients.

Variables	Healthy Controls	ALS Patients	*p* Value
Participants (number)	34	47	
Age (years)	54 ± 7	58 ± 10	0.1161
Sex (female/male)	14/20	15/32	
Site of onset			
Bulbar	NA	13	
Limb	NA	34	
BMI (kg/m^2^)	23.53 ± 2.172	23.28 ± 2.463	0.7523
Disease duration (months)	NA	10.15 ± 5.564	
CRP (mg/L)	NA	2.369 ± 2.438	
IL-6 (pg/mL)	NA	7.098 ± 7.962	
RABGGTB	5.671 ± 2.932	17.34 ± 8.226	<0.0001
ALSFRS-R score	NA	39.38 ± 4.465	
ΔFS	NA	1.119 ± 0.997	

BMI, body mass index; CRP, C-reactive protein; IL-6, interleukin 6; RABGGTB, Rab geranylgeranyltransferase subunit beta; ALSFRS-R, ALS functional rating scale—revised; ALS, amyotrophic lateral sclerosis; ΔFS, delta FS ratio (rate of disease progression) = (48 – ALSFRS-R score at time of diagnosis)/time onset to diagnosis); NA, not available.

## Data Availability

The datasets generated during and/or analyzed during the current study are available from the corresponding author upon reasonable request.

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
