# Peer review of "Rab Geranylgeranyltransferase Subunit Beta as a Potential Indicator to Assess the Progression of Amyotrophic Lateral Sclerosis"

_brainsci, 2023, doi:10.3390/brainsci13111531_

Round 1

Reviewer 1 Report

Comments and Suggestions for Authors

The authors had explored on “RABGGTB as a potential indicator in evaluating amyotrophic lateral sclerosis progression”.  The research is very extensive, well structured and defined throughout the article. Also, provide some information to support your points for enhancing your article’s impact.

COMMENTS:

1.     Consider revising the title to make it more descriptive that highlights the most aspects of your work.

2.     The introduction should clearly state the objectives of the study and consider providing a more comprehensive context of the research

3.     In the introduction, the statement “However, there has been no consensus” is indefinite. Elaborate on what consensus is lacking and its significance on ALS.

4.     Provide information on the current state of research in ALS treatment and biomarker identification.

5.     Discuss more about other proteins than  RAB or factors that involved in ALS development.

6.     Provide a detailed description of the methodology used for collecting and processing serum samples.

7.     In methodology section of  flow cytometry, give details on the concentration of primary antibody.

8.     Discuss the limitations of using RABGGTB as a potential indicator in evaluating amyotrophic lateral sclerosis progression.

9.     In the result section, eloborate on range of AUC value and its representation of the area under the ROC curve.

10.  Provide the article citation related to previous evidence studies and suggetsion.

11.  Provide information on the other prospects of Amylotrophic lateral disease progression and prognosis.

12.  Ensure, is there any previous studies suggesting the significant correlation between age and other variables for ALS?

13.  Elaborate on how BMI is associated with the incidence of ALS other than high catabolism.

14.  Are the conclusion supported by the results?

Reviewer 2 Report

Comments and Suggestions for Authors

The authors show RABGGTB as a new biomarker for ALS, useful not only for the diagnosis of motor neuron disease but also as a marker of the rate of disease progression.

A few points need to be addressed by the authors:

1) Why did the authors not check the biomarker neurofilaments heavy and light chain in correlation with the disease score and progression rate?

2) It is still unclear, if the biomarker RABGGTB is changed very early in the course of ALS.

3) The rate of bulbar-onset ALS is quite high, why?

4) Please provide more information to the macrophage markers CD14 and CD 16. Are functions of these cell surface markers known?

5) Figure 2 is not labeled correctly in the text.

6) Please provide further explanations what is shown in Figure 4. The term ROC requires further explanation.

7) With regard to the first page of the Discussion: Microglial cells do not really play a crucial role in the onset of ALS. They are rather important for the progression of the disease, please see selective silencing data from Cleveland group.

8) Is there any correlation between the expression of CD14 and CD16 on macrophages/microglia and the M1 or M2 phenotype of these cells?

9) Line 303: survival, not sur vival.

10) What are the possible pathophysiological consequences of the higher expression of RABGGTB in macrophages? There is no discussion on this point. Is the higer expression a primary phenomenon in macrophages or just secondary to the inflammatory state of the cells?

Comments on the Quality of English Language

good and clear

Reviewer 3 Report

Comments and Suggestions for Authors

Based on unpublished preliminary data this work intends to present a new marker for the devastating disease of ALS. The marker RABGGTB is however not new for other diseases such as MS or cancer. This immediately opens the question of the specificity and selectivity of the marker. 

On the other hand, the authors show that this marker could be a good indicator of ALS progression. However, this finding of correlations is hampered by the lack of some expected correlations such as age, disease course. The latter is circumvented by stating "limited sampling regions and small sample size". This raises the question: wouldn't this also affect the significance of positive correlations?

In addition, the authors compared some other known molecules - where data exist that these could also serve as markers in ALS - CRP and IL-6 and could not demonstrate a correlation. However, they did not test/compare some hardcore blood markers such as neurofilaments or TDP43 or C9ORF72 or others...

Finally, in Discussion it is emphasized that the expression of RABGGTB is essential for clinical diagnosis of ALS. However later on we learn that this is best for fast progressing disease types. Could the authors then state what is the diagnostic significance of this marker for other disease groups showing a lower AUC value?

Minor comments

It is never explained why did the PMBCs have to be differentiated in vitro into macrophages prior to testing for RABGGTB and not tested directly.

Line 159 - healthy controls are "subjects" not "patients".

Line 160  - what does it mean to have a mean age on two or even tree decimal places?! In addition decimal places of the mean value and error should match (and relate to precision of measurement) - note also Tab. 1

Fig, 2 -panels are wrongly references in the text.

Lines 276-277 - why are microglia dicussed when these cellsa re not studied?

Line 288  - hSOD1-

There is extensive Discussion on BMI and IL-6 but these are shown not to be correlated to RABGGTB. On the other hand, the lack of association of RABGGTB and even ALSFRS - R score with disease progression should be discussed.
